# Changes in Tumor Necrosis Factor α (TNFα) and Peptidyl Arginine Deiminase 4 (PAD-4) Levels in Serum of General Treated Psoriatic Patients

**DOI:** 10.3390/ijerph19148723

**Published:** 2022-07-18

**Authors:** Joanna Czerwińska, Marta Kasprowicz-Furmańczyk, Waldemar Placek, Agnieszka Owczarczyk-Saczonek

**Affiliations:** Department of Dermatology, Sexually Transmitted Diseases and Clinical Immunology, School of Medicine, University of Warmia and Mazury in Olsztyn, 10-719 Olsztyn, Poland; martak03@wp.pl (M.K.-F.); w.placek@wp.pl (W.P.); aganek@wp.pl (A.O.-S.)

**Keywords:** PAD-4, NETs, psoriasis, systemic therapy, adalimumab, secukinumab, methotrexate

## Abstract

Psoriasis is an autoimmune disease in which the disturbed dependencies between lymphocytes, dendritic cells, keratinocytes and neutrophils play the most important role. One of them is the overproduction of neutrophil extracellular traps (NETs). The release of NETs can be induced by pathogens, as well as antibodies and immune complexes, cytokines and chemokines, including TNFα. The first step of the NET creation is the activation of peptidyl arginine deiminase 4 (PAD-4). PAD-4 seems to be responsible for citrullination of histones and chromatin decondensation, but the data on PAD-4 in NETs is inconclusive. Thus, the current study aimed to determine PAD-4 and TNFα levels in the serum of psoriatic patients by ELISA and observe the response of these factors to systemic (anti-17a, anti-TNFα and methotrexate) therapies. Increased levels of both PAD-4 and its main stimulus factor TNFα in pre-treatment patients have been reported along with the concentrations of proteins correlated with disease severity (PASI, BSA). Before treatment, the irregularities in the case of anti-nuclear antibodies level (ANA) were also observed. All of the applied therapies led to a decrease in PAD-4 and TNFα levels after 12 weeks. The most significant changes, both in protein concentrations as well as in scale scores, were noted with anti-TNFα therapy (adalimumab and infliximab). This phenomenon may be associated with the inhibition of TNFα production at different stages of psoriasis development, including NET creation. The obtained data suggest the participation of PAD-4 in the activation of neutrophils to produce NETs in psoriasis, which may create opportunities for modern therapies with PAD inhibitors. However, further exploration of gene and protein expression in psoriatic skin is needed.

## 1. Introduction

Psoriasis is a chronic inflammatory systemic disease with the burden of comorbidities, i.a., respiratory, cardiovascular and gastrointestinal [1,2,3]. This disorder is a serious global problem, which concerns at least 4.5 million individuals, according to the Global Burden of Disease (2019). The average age of patients with psoriasis is about 58 years (without differences between sexes). The most incident cases of psoriasis were noted in the most developed countries of Europe and North America [4]. Psoriasis significantly worsens the quality of life of patients and their families, often causing them to withdraw from social life. Despite the fact that the symptoms of psoriasis are typical and the diagnosis is usually not problematic, it is still a serious therapeutic challenge, mainly due to the complicated mechanism of its development. Every year, new elements of the immune system that may affect the pathogenesis of psoriasis are discovered. The most important role is played by the autoinflammatory feedback loop between lymphocytes, dendritic cells and keratinocytes, which activate the control axis (TNFα/IL-17/IL-23) to produce pro-inflammatory interleukins [3,5,6,7]. Additionally, neutrophils, which have the ability to create extracellular traps (NETs) abundant in the cytoplasmic granules and DNA, strongly affect this interaction. Three typical stages may be distinguished in the NET formation process: neutrophil activation, changes in cell and extracellular protein, and DNA release. Neutrophils are induced by bacteria, fungi, and parasites [8]. The factors described as stimulating NETs include also antibodies and immune complexes, cytokines, and chemokines [9]. The main representative of the immune group is TNFα—the major stimuli factor of the protein arginine deiminase 4 (PAD-4), i.e., enzyme responsible for the onset of changes in the cell starting from chromatin decondensation (deamination/citrullination) [8]. In activated neutrophils, PAD-4 is translocated to the nucleus, wherein it is stimulated by Ca2+ levels above the physiological concentration [10]. It has been shown that three key molecules are engaged in the phenomena inside the cell: neutrophil elastase (NE) responsible for histone degradation, myeloperoxidase (MPO) changing chromatin structure, and cathelicidin antimicrobial peptide (LL-37) connecting histones with DNA. In the intracellular space, there is also increased ROS generation caused by the activation of nicotinamide adenine dinucleotide phosphate (NADPH) oxidase. The lack of a positive charge of arginine leads to the breaking of nucleosomes and nuclear envelope, and to hampered degranulation. The last stage entails the release of the chain of DNA and protein (e.g., MPO, NE and LL-37) into the extracellular space [5,6,7] (Figure 1).

In physiological conditions, NETs dispose of various pathogens, acting as the first line of the body’s defenses by direct trapping microorganisms [7,8,11,12]. However, the dysregulated NET activity may lead to aggravated inflammation, impairment of microcirculation, and cause tissue damage and then trigger autoimmune diseases, such as psoriasis [13,14]. The key role is played by LL-37 that creates complexes with DNA/RNA and causes Th1/Th17/Th22 differentiation. Additionally, the pathogenesis of psoriasis follows two more mechanisms: plasmacytoid dendritic cell (pDC) activation via Toll-like receptors (TLRs) and interferon-α (INF-α) release [15,16,17], which confirms the complex and complicated mechanism of this disease development (Figure 1).Thus, the NETs are found at each stage of psoriasis development, where they are engaged in an immune response manifested by T-cell imbalance, keratinocyte hyperproliferation, and the formation of autoantigens [18,19] (Figure 1).

The immune system is able to produce, for example, anti-neutrophil cytoplasmic antibodies (ANCA), directed against proteins present in neutrophil granules found in systemic small vessel vasculitis (SVV) or anti-nuclear antibodies (ANA, i.a. anti-dsDNA/ssDNA, anti-histone) against nucleus components, which are characteristic for systemic lupus erythematosus (SLE) and they are very often found in psoriasis [20,21]. Hence, autoantibodies accumulate in the tissues in the form of immune complexes (e.g., together with platelets, erythrocytes and plasma proteins), initiating cytotoxic reactions [11] that cause modifications of host proteins, including their citrullination and the production of anti-citrullinated protein antibodies (ACPA). In turn, it leads to the loss of immunological tolerance to these proteins [11,12,13,14,15,16,17,18,19,20]. Additionally, it was noted that patients with autoimmune diseases have a limited ability to degrade NETs by decreased activity of DNase, which is related to the increased presence of its inhibitors [21]. Moreover, DNase is able to remove DNA (NET’s backbone), but not the proteins attached. This is due to the fact that NET components bind secondarily to proteins present on the host vascular endothelium [22], which contributes to damage to its own tissues and the development of diseases [23]. The excessive NET formation connecting with the externalization of citrullinated autoantigens (vimentin, α-enolase), pro-inflammatory cytokines (TNFα, interleukin IL-6, IL-8), and immunostimulatory molecules impair immune response by leading to inflammation [24,25].

The course of psoriasis is usually characterized by a high accumulation of neutrophils, both in plaques as well as in the blood [26,27,28]. The cells circulating with blood, low-density granulocytes (LDGs) with ability to synthesize TNFα, are the most essential to NET production [20] in psoriatic patients. Interestingly, the decreased amount of NETotic cells in the blood correlates with the regression of psoriasis [7,29,30]. Moreover, the sera from psoriatic patients, added to healthy neutrophils, are able to induce NET production [7]. Additionally, biological therapies may indirectly affect the function and number of active neutrophils [30]. It is worth noting that both severe lymphocyte infiltration in psoriatic skin lesions, as well as subcorneal Munro’s microabscesses filled with neutrophils, are characteristic of the pathological image of the diseases [8].

Interestingly, disruptions in NET production caused by irregularities in the activity of the PAD family (especially PAD-1 and 2) are also indicated. Isoforms are clearly differentiated in terms of occurrence and functions, e.g., PAD-1 occurs in the epidermis, whereas PAD-2 is expressed in various organs, i.a., in the brain, female reproductive tissues, and skeletal muscle, but it is also found in the periphery of granular keratinocytes. Hair follicles and epithelium exhibit PAD-3 expression, and PAD-6 is found in germ cells. Hence, PAD-1 and PAD-3 are engaged in epidermal homeostasis and barrier function, whereas PAD2 is responsible for cytoskeletal degeneration and apoptosis. In turn, PAD-6 regulates mainly female infertility [24,25]. In turn, the highest stability and activity is presented by PAD-4, whose expression is restricted to immune cells, such as neutrophils, eosinophils, granulocytes, and monocytes, where it participates in the regulation of inflammatory processes [24].

Despite the fact that numerous studies have addressed different disorders (mainly rheumatoid arthritis-RA and Bechet’s disease-BD), data concerning the necessity of PAD-4 involvement in NET generation still remain contradictory. Both reports showing that the enzyme is essential to NET generation as well as works rejecting this hypothesis may be found in the literature [31,32,33,34,35]. Thus, the role of PAD-4 in the transformation of chromatin structure in neutrophils remains unclear. Moreover, there is still a paucity of research on the involvement of this enzyme in the formation of NETs in human psoriasis. Therefore, the aim of the current study was to determine the levels of PAD-4 and its main activator (TNFα) in the serum of psoriatic patients and observe the response of these factors to systemic therapy: anti-IL-17A (ixekizumab, secukinumab), anti-TNFα (adalimumab, infliximab), and methotrexate (MTX), mostly deployed in psoriasis treatment.

## 2. Material and Methods

### 2.1. Study Group

The study group included 50 patients aged 18–65 without significant burdens (Table 1) and treated in the Clinic and at the Department of Dermatology, Sexually Transmitted Diseases and Clinical Immunology in Olsztyn, for plaque psoriasis. Patients with other inflammatory diseases, neoplastic diseases, previous cardiovascular complications, heart, kidney and liver failure were excluded. The control group consisted of healthy volunteers (*n* = 16) with no personal and family history of autoimmune and inflammatory diseases. Patients were classified into three research groups. The first group received methotrexate (MTX): orally at a dose of 15 mg/week. The second therapy included the administration of anti-TNF (adalimumab): subcutaneously, initial dose 80 mg, after seven days, 40 mg every two weeks. The third group took an IL-17a inhibitor (secukinumab): subcutaneously, 300 mg at 0–4 weeks, then 300 mg every month. Skin condition assessment (Psoriasis Area Severity Index–PASI, Body Surface Area (BSA), and Dermatology Life Quality Index (DLQI)) were performed on each visit and always by the same qualified person.

### 2.2. Methods

Before the start of treatment (week 0), the level of neutrophils, C Reactive Protein (CRP) and titer of anti-nuclear antibodies (ANA, Euroimmun, Lübeck, Germany) were detected. Hematology and biochemical analyses of blood were performed in the Laboratory of the Municipal Polyclinical Hospital in Olsztyn. All ANA analyses were followed by an Immunoblot test (ANA profile V and ANA Profile 23, Euroimmun, Germany). PAD-4 and TNFα levels in blood serum were assessed by the enzyme-linked immunosorbent assay (ELISA) using commercially available kits (Cayman Chemical, MI, USA; EIAAB, Wuhan, China) with a wide detection range: 156–10,000 pg/mL (PAD-4) and 15.6–1000 pg/mL (TNFα). The procedures were performed according to the manufacturer’s protocol. The assay’s validity was confirmed based on parallelism between the standard curve and dilutions of randomly chosen serum samples. The intra- and inter-assay coefficient of variation for factors was <4%. The sensitivity of these assays was defined as the least concentration that could be differentiated from zero samples, and it was determined at 78 and 7.8 pg/mL for PAD-4 and TNFα, respectively. Absorbance values were measured at 450 nm using Multiskan FC (Thermo Fisher Scientific, CO, USA).

### 2.3. Statistical Analysis

The results were processed statistically by the following non-parametric tests: the Mann–Whitney U test (comparison of two independent groups), Friedman’s ANOVA (comparison of many dependent groups) and Spearman’s correlation test. The results were expressed as mean ± SEM. All calculations were performed using Statistica software, version 13.1 (Statsoft, Inc., Tulsa, OK, USA). Differences were regarded as statistically significant at *p* < 0.05.

## 3. Results

In the presented study, the concentrations of NET protein–PAD-4 and its main stimulus factor, TNFα, were analyzed before and during (week 4 and 12) the application of three types of systemic therapy used in patients with psoriasis: MTX, anti-TNFα and anti-IL 17a.

### 3.1. Neutrophils, C Reactive Protein (CRP) and Anti-Nuclear Antibodies (ANA)

Before starting the treatment, the level of neutrophils, C Reactive Protein (CRP) and the titer of anti-nuclear antibodies (ANA) were detected (Table 2). Interestingly, in all of the patients, the level of neutrophils, as well as CRP, were normal, whereas the titer of ANA was increased. All qualified patients showed a titer level higher than 1:320. The ANA pattern was defined as granular or mixed. An analysis of ANA was followed by an immunoblot test (ANA Profile V and ANA Profile 23). The results were mostly negative, and the type of antibodies was not detected. However, three patients indicated positive mitochondria (AMA), double-stranded DNA (dsDNA) and an Ro-52 antigen band. After 12 weeks of treatment, the levels of described factors did not change significantly. However, the use of anti-TNFα antibodies elevated the titers of ANA by at least one level (1:640–1:250) in most patients. In other groups (MTX and anti-Il 17a), the titer of ANA changed in only 10% of patients.

### 3.2. Psoriasis Area Severity Index (PASI), Body Surface Area (BSA), and Dermatology Life Quality Index (DLQI)

On each measurement point, the Psoriasis Area Severity Index (PASI), Body Surface Area (BSA) and Dermatology Life Quality Index (DLQI) were evaluated. Before therapy, 38 patients indicated a PASI score of more than 10 and a BSA score of more than 16, whereas all of them (*n* = 50) had a DLQI higher than 10 (Table 3).

During the study, scores on all scales (PASI, BSA, DLQI) decreased (Table 4). The best results were observed for therapy with anti-TNFα (Table 5), where the skin condition assessment indicated changes of more than 65% (*p* < 0.05).

### 3.3. Peptidyl Arginine Deiminase 4 (PAD-4)

Among a full group of patients (*n* = 50), there was a correlation between PAD-4 with PASI score (r = 0.3141, *p* < 0.05). The analyses of protein levels also showed a correlation between PAD-4 and BSA (r = 0.6401, *p* < 0.05). However, no correlation was found between the duration of the disease (in years) and the concentration of the studied factor before the start of therapy (r = 0.2218). The concentration of PAD-4 was significantly higher in the group of PASI > 10 and BSA > 16 in comparison with PASI ≤ 10 and BSA ≤ 16. The concentration of PAD-4 in psoriatic patients was also higher than in healthy volunteers (*p* < 0.05) (Table 6).

The analysis of the entire group of patients (*n* = 50) showed a decrease in concentrations (*p* < 0.05) of the tested factor after four weeks of treatment. The level of PAD-4 remained at a similar level also after 12 weeks (Table 7).

The division of the research group in relation to the therapy allowed for the analysis of changes in the concentrations of the tested factor during the course of each of them. It was observed that the level of PAD-4–the enzyme protein initiating the formation of NET –decreased (*p* < 0.05) as a result of treatment with all three therapies (Figure 2, Table 8). However, in the case of treatment with therapy 2 (anti-TNFα), the greatest decrease in PAD-4 concentration was observed (81%, *p* < 0.05). The protein concentration was lower than in the group of healthy volunteers (*p* < 0.05). In the remaining cases (therapy 1 and 3), the concentration of the PAD-4 after 12 weeks of treatment was still higher (*p* < 0.05) or equal in comparison with the control group. The smallest differences were observed in therapy with anti-IL17a (therapy 3).

### 3.4. Tumor Necrosis Factor α (TNFα)

Similar to PAD-4, the concentration of TNFα was also correlated with PASI and BSA (r = 0.3668 and r = 0.4142, respectively, *p* < 0.05). Thus, the correlation between TNFα and PAD-4 was noted (0.4061, *p* < 0.05). The level of TNFα was elevated in moderate and severe psoriasis (PASI > 10 and BSA > 16) compared with mild forms of the disease (Table 9). The concentration of this factor in psoriatic patients was also significantly higher than in the control group (*p* < 0.05).

In the entire population with the disease (*n* = 50), TNFα level decreased during treatment (*p* < 0.05). The observed changes were comparable to that of PAD-4 (Table 10).

The analyses of protein levels in different research groups (therapy 1–3) indicated a significant decrease in only two applied therapies: MTX and anti-TNFα. However, a larger change was noted with anti-TNFα treatment (78%, *p* < 0.05). Additionally, the TNFα level after 12 weeks of using different therapies (1–3) still remained higher (*p* < 0.05) than in the group of healthy volunteers (Figure 3, Table 11).

## 4. Discussion

The results of the current study provide novel evidence for the presence of PAD-4 protein in human serum. To the best of the authors’ knowledge, this is the first study demonstrating the level of this enzyme in psoriatic patients. The PAD-4 and TNFα responses to three types of systemic treatment were also analyzed. Before treatment, neutrophils and CRF levels, as well as ANA titer, were evaluated. The irregularities were observed only in the case of ANA. Additionally, after 12 weeks of therapy with anti-TNFα, elevated ANA titers were noted in most patients but without reference to clinical symptoms. Similar observations were presented in psoriatic patients by Yasuda et al. (2019) [36].

The study was performed at three different measurement points (before the study and in the 4th and 12th week of therapy). The study confirmed the role of TNFα as an indicator of the disease severity [37], but it also determined, for the first time, that this role may be played by PAD-4. Similar to TNFα, its concentration also correlated with the PASI and BSA score. Moreover, the levels of both TNFα and PAD-4 were significantly increased in psoriatic patients compared with the control group.

The study found that the levels of tested factors varied under the influence of all treatments. TNFα concentration was best influenced by its inhibitors: adalimumab and infliximab. Likewise, the greatest decrease in PAD-4 concentration was observed under general therapy with anti-TNFα. The PASI, BSA and DLQI scores confirmed these observations. The spectacular improvement of the indices was observed in the group treated with biological therapy with anti-TNFα. This phenomenon may be associated with a broad spectrum of action of therapies and inhibition of TNFα production at three stages of psoriatic pathogenesis: from activated LDGs, cDC and Th1 cells (Figure 1). It also confirmed the participation of NET in psoriasis development. Interestingly, in contrast to the current study, a review by Bergen (2020) found treatment with inhibitors of IL-17 to be the most clinically beneficial [32].

In the available literature, there are few studies analyzing PAD-4 protein concentrations in human serum. Thus, the discussion of the current results is difficult. To date, anti-PAD-4 antibodies have been analyzed in the blood of patients with RA. It has been shown that antibodies against PAD-4 precede the clinical onset of RA in a significant proportion of patients and may be important in the initial stage of the disease [38,39], which indirectly confirms the current results. In turn, studies with patients with active BD exhibit higher mRNA PAD4 expression than in inactive BD and healthy volunteer groups. Additionally, neutrophils of healthy subjects exposed to sera from BD patients induced NADPH oxidase protein expression and affected NET formation [40]. PAD-4 is also involved in skin tumorigenesis, in which the inhibition of this enzyme results in apoptosis as well as in the repression of p53 target genes [28]. The main role of the stimuli factor of NETs in autoimmune diseases is assigned to the TNFα/IL-17/IL-23 axis [10,11,12]. These signals cause the activation of the NADPH oxidase and thus the increase in ROS. Interestingly, there are also reports indicating pathways independent of NADPH [41,42]. It is suggested that only in this case are the participation of PAD-4 and chromatin decondensation needed for the NET formation [38]. The studies performed by Doud et al. (2013) and Neeli et al. (2013) excluded the absolute necessity of the PAD-4 enzyme in the initiation of NADPH-dependent NET. In the described experiments, stimulation of NET with phorbol myristate acetate (PMA) led to the formation of traps without detectable histone deimination [41,42]. Additionally, the use of PMA, which is also an activator of protein kinase C (PKC)–an inhibitor of PAD-4, showed no relationship between the NET formation and histone citrullination [43]. However, contrary evidence indicating that the NET formation requires PAD-4 activation was presented. Research on PAD-4 was mainly conducted using the mouse animal model. It was reported that histone citrullination correlated with chromatin decondensation during the NET formation. Blocking PAD-4 was found to disrupt network formation and therefore increase the risk of bacterial infection in animals [26]. In addition, the study of Ding et al. (2022) showed remarkably ameliorated psoriasis-like symptoms in PAD-4 knockout mice [44]. Thus, studies using knock-out animals indicated the participation of PAD-4 as a required component for NET-dependent innate antimicrobial immunity [26] as well as a factor firmly associated with psoriasis development [44]. Additionally, PAD-4 has been shown to participate in the regulation of proliferation and hematopoiesis [42]. Studies using the HL-60 cell line clearly indicated that PAD-4 citrullination of histones mediates chromatin decondensation and the inhibition of PAD-4 reduces histone hypercitrullination and the formation of NET structures [25,45,46]. The cellular events that lead to the NET formation were presented by Thiam et al. (2020) for human and mouse primary blood neutrophils, as well as in differentiated HL-60 neutrophil-like cells (dHL-60). Those authors showed that extracellular DNA release required the enzymatic and nuclear localization activities of PAD-4 [46]. Studies using the knock-out mouse model of RA showed reduced severity of symptoms by inducing immune reactions such as helper T-cell development, inflammatory cytokine production and apoptosis [47,48]. In addition to RA, the autoimmune disease in which PAD-4 was studied was also lupus (LE). Again, using a mouse animal model and PAD-4 inhibitors, a reduction in the autoimmune response was observed [49]. In sick mice, the inhibitors of PAD suppressed the formation of NETs, reduced the accumulation of immune complexes, i.a., in the kidneys and improved the condition of the skin [50]. In addition, it has been shown that PAD-4 inhibition reduces NET-related citrulline histones, and minimizes immune cell recruitment [51] and liver damage associated with ischemia-reperfusion injury [52]. In contrast, PAD-4 from bone-marrow-derived cells and NETs did not influence chronic atherogenesis, but its involvement in acute thrombotic complications of intimal lesions was noted. Thus, the contribution of NETs and PAD-4 to atherosclerosis and its thrombotic complications seems to be stage- specific [53].

In the newest reports, the hypothesis that abnormality in NET creation is strongly associated with psoriasis development is discussed more and more frequently. The high levels of NETs in the lesions of psoriatic patients are still described [54,55,56]. However, huge attention is also paid to irregularities in PAD isoform activity. Interestingly, a new direction of PAD analyses was proposed by Bawadeker et al. in 2017. The authors induced the NET formation in mice with TNFα-induced arthritis and noted the participation of PAD-2 in protein citrullination [57]. In turn, Padhi et al. (2022) have recently reported the correlation between Il-22+cells and PAD-1 in lesional psoriatic skin, where IL-22 inhibited the expression of PAD-1 in epidermal keratinocytes [58]. Thus, the lack of citrullinated by PAD-1 keratin K1 in skin samples from psoriasis patients may be a contributing factor to disease [54]. The results of both studies confirmed the important role of PAD isoforms as significant elements regulating cytokine secretion in the control axes, which may be crucial in the mechanism of various diseases.

Concurrently, we may find the reports providing information about additional mechanisms engaged in the NET generation. Meng (et al. 2022) noticed that decreased receptor-interacting serine/threonine-protein kinase 1 (RIPK1) expression in psoriasis neutrophils may enhance traps [59], whereas Ding et al. (2022) indicated that Src homology-2 domain-containing protein tyrosine phosphatase-2 (SHP2) had a high impact on psoriasis severity [44].

The presented data indicated different pathogenetic mechanisms of autoimmune diseases that involve both PAD-dependent and PAD-independent pathways. Simultaneously, it appeared to be an extensive field for NET research because knowledge of the contribution of PAD-4 to the formation of NET networks in psoriasis is still scarce. The current results, showing a clear decrease in PAD-4 concentration in serum, may indicate the enzyme’s involvement in the activation of NETs and psoriasis development. However, fully defining PAD-4 as an indicator of psoriasis severity requires additional analyses at both the protein and the gene levels in the skin.

Due to the fact that PAD-4 shows the highest stability and activity in inflammation and it occurs in distinctive cells, this enzyme seems to be particularly dangerous in the pathogenesis of skin barrier defects, including psoriasis. Hence, the PAD-4 may be a primary target for modern therapies with PAD-4-specific inhibitors. These therapies may impede the functioning of the feedback loop between neutrophils (by inhibiting the NET formation without damaging beneficial defensive properties), dendritic cells, lymphocytes and keratinocytes. A reduction in disease symptoms using PAD-inhibitors has been noted in the mouse models of RA and LE. Interestingly, the current results also showed that cytokine-targeted therapies indirectly altered PAD-4 levels. However, further research is still needed regarding PAD activation as a main component of NET in the pathogenesis of psoriasis.

## Figures and Tables

**Figure 1 ijerph-19-08723-f001:**
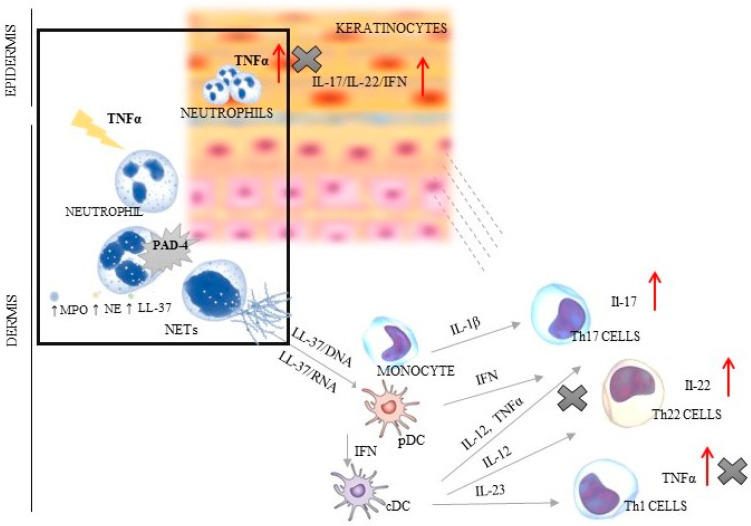
The dependence between NET creation and TNF/IL-23/IL-17 axis in psoriatic skin (the box shows the relation of TNFα and PAD-4; ↑ indicates an increase in the secretion of presented factors; X indicates the sites of inhibition of TNFα production by using of anti-TNF therapy). PAD-4—peptidyl arginine deiminase 4; NE—neutrophil elastase; MPO—myeloperoxidase; cDC—conventional dendritic cell; IFN—interferon; IL—interleukin; LL37—cathelicidin; NETs—neutrophil extracellular traps; pDC—plasmacytoid dendritic cells; Th—T helper lymphocyte; TNF—tumor necrosis factor.

**Figure 2 ijerph-19-08723-f002:**
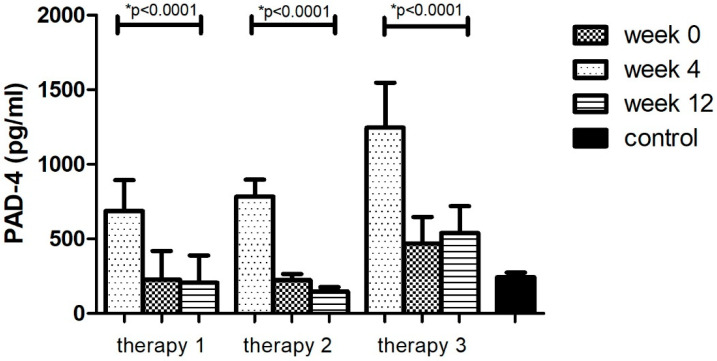
PAD-4 level (pg/mL) before (week 0) and during (week 4, week 12) therapy with: MTX (therapy 1, *n* = 20), anti-TNFα (therapy 2, *n* = 20) and anti-IL-17A (therapy 3, *n* = 10) in patients with plaque psoriasis compared with a control consisting of healthy volunteers (*n* = 16). * indicates a statistically significant difference (*p* < 0.05) between weeks: 0 and 12.

**Figure 3 ijerph-19-08723-f003:**
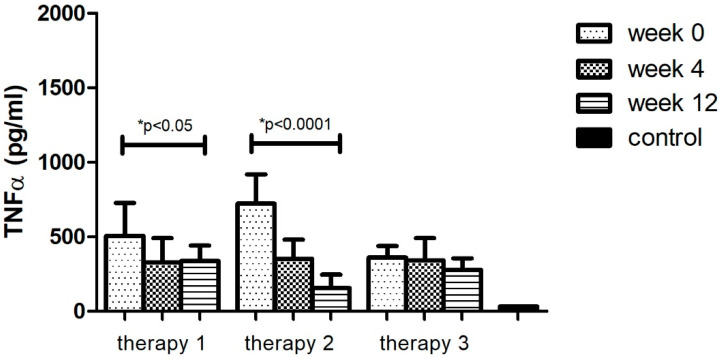
TNFα level (pg/mL) before (week 0) and during (week 4, week 12) therapy with: MTX (therapy 1, *n* = 20), anti-TNFα (therapy 2, *n* = 20), anti-IL-17A (therapy 3, *n* = 10) in patients with plaque psoriasis compared with a control consisting of healthy volunteers (*n* = 16). * indicates a statistically significant difference (*p* < 0.05) between weeks: 0 and 12.

**Table 1 ijerph-19-08723-t001:** Clinical characteristics of patients with psoriasis.

Variables	Results	Range
**patients**	50 (39M, 11F)	
**age (years)**	52.64 ± 2.1	18–65
**illness duration (years)**	17.21 ± 2.3	2–36

**Table 2 ijerph-19-08723-t002:** Laboratory analysis of patients’ serum before starting the therapy.

Variables	Results	Range
neutrophils (%)	57.74 ± 2.5	40–70
neutrophils (10^3^/μL)	4.23 ± 0.3	-
CRP	2.58 ± 0.6	<0.5
ANA	1:320–1:1280	<1:80

**Table 3 ijerph-19-08723-t003:** Levels of PASI, BSA and DLQI indicators before the study.

Variables	Week 0	Mean
PASI	<10 (*n* = 12)	8.13 ± 0.6
	>10 (*n* = 38)	18.31 ± 0.9
BSA	<16 (*n* = 12)	10.46 ± 0.8
	>16 (*n* = 38)	30.72 ± 4.0
DLQI	>10 (*n* = 50)	19.97 ± 1.4

**Table 4 ijerph-19-08723-t004:** Levels of PASI, BSA and DLQI indicators before the study (week 0) and during treatment (week 12). * indicates a statistically significant difference (*p* < 0.05).

	Week 0	Week 12	Effect (↓)
PASI	15.74 ± 1.6	5.93 ± 1.0	* 62%
BSA	24.03 ± 4.1	8.15 ± 2.2	* 66%
DLQI	20.09 ± 7.6	8.84 ± 1.2	* 55%

**Table 5 ijerph-19-08723-t005:** The comparison of PASI, BSA and DLQI scores between therapies with: MTX (therapy 1, *n* = 20), anti-TNFα (therapy 2, *n* = 20), anti-IL17a (therapy 3, *n* = 10) in patients with plaque psoriasis at two measurement points (week 0, week 12). * indicates a statistically significant difference (*p* < 0.05).

**Therapy 1**		**Week 0**	**Week 12**	**Effect (** **↓** **)**
	PASI	12.11 ± 1.1	5.01 ± 0.8	* 58%
	BSA	9.28 ± 3.2	7.84 ± 1.3	15%
	DLQI	22.86 ± 4.3	11.87 ± 1.1	48%
**Therapy 2**	PASI	19.83 ± 1.4	4.49 ± 0.9	* 77%
	BSA	42.23 ± 5.4	10.41 ± 2.4	* 75%
	DLQI	19.03 ± 1.2	6.16 ± 1.2	* 67%
**Therapy 3**	PASI	15.3 ± 2.1	8.29 ± 1.3	45%
	BSA	20.60 ± 3.8	6.20 ± 2.9	* 69%
	DLQI	18.40 ± 2.1	8.50 ± 1.1	* 53%

**Table 6 ijerph-19-08723-t006:** Pre-treatment levels of PAD-4 (pg/mL) in patients with plaque psoriasis (*n* = 50) with PASI score ≤ 10 (*n* = 12), PASI > 10 (*n* = 38), BSA ≤ 16 (*n* = 12), BSA > 16 (*n* = 38) and in the control group (*n* = 16).

Variable	Group	Mean (pg/mL)
PAD-4	PASI ≤ 10	391.78 ± 26.3
	PASI > 10	1019.35 ± 62.3
	BSA ≤ 16	361.97 + 20.1
	BSA > 16	1022.31 ± 60.7
	control	244.88 ± 28.8

**Table 7 ijerph-19-08723-t007:** Changes of PAD-4 level (pg/mL) during treatment (*n* = 50). * indicates a statistically significant difference (*p* < 0.05).

Variable (pg/mL)	Week 0	Week 4	Effect (↓)	Week 12	Effect (↓)
PAD-4	905.26 ± 29.9	344.91 ± 21.7	* 61%	297.82 ± 21.3	* 67%

**Table 8 ijerph-19-08723-t008:** PAD-4 level (pg/mL) between treatments: MTX (therapy 1, *n* = 20), anti-TNFα (therapy 2, *n* = 20) and anti-IL-17A (therapy 3, *n* = 10) in patients with plaque psoriasis at three measurement points (week 0, week 4, week 12). * indicates a statistically significant difference (*p* < 0.05).

PAD-4	Week 0	Week 4	Effect (↓)	Week 12	Effect (↓)
Therapy 1	686.92 ± 20.8	225.61 ± 19.3	* 67%	206.52 ± 18.2	* 69%
Therapy 2	783.89 ± 11.34	223.26 ± 4.1	* 71%	148.1 ± 2.8	* 81%
Therapy 3	1245.1 ± 57.6	568.0 ± 45.47	* 54%	539.0 ± 45.3	* 56%

**Table 9 ijerph-19-08723-t009:** Pre-treatment levels of PAD-4 (pg/mL) in patients with plaque psoriasis (*n* = 50) with PASI score ≤ 10 (*n* = 12), PASI > 10 (*n* = 38), BSA ≤ 16 (*n* = 12), BSA > 16 (*n* = 38) and in the control group (*n* = 16).

Variable	Group	Mean (pg/ml)
TNFα	PASI ≤ 10	450.81 ± 325.9
	PASI > 10	598.8 ± 121.61
	BSA ≤ 16	500.1 ± 272.6
	BSA > 16	605.76 ± 130.2
	control	32.15 ± 1.99

**Table 10 ijerph-19-08723-t010:** Changes of TNFα level (pg/mL) during treatment (*n* = 50). * indicates a statistically significant difference (*p* < 0.05).

Variable (pg/mL)	Week 0	Week 4	Effect (↓)	Week 12	Effect (↓)
TNFα	529.72 ± 164.9	341.37 ± 146.1	35%	256.71 ± 90.7	* 51%

**Table 11 ijerph-19-08723-t011:** TNFα level (pg/mL) between treatments with: MTX (therapy 1, *n* = 20), anti-TNFα (therapy 2, *n* = 20) and anti-IL-17A (therapy 3, *n* = 10) in patients with plaque psoriasis at three measurement points (week 0, week 4, week 12). * indicates a statistically significant difference (*p* < 0.05).

TNFα	Week 0	Week 4	Effect (↓)	Week 12	Effect (↓)
Therapy 1	505.26 ± 223.1	329.19 ± 161.8	34%	336.77 ± 103.9	* 33%
Therapy 2	722.34 ± 196.1	352.92 ± 127.5	* 51%	155.07 ± 91.5	* 78%
Therapy 3	361.57 ± 75.6	342.0 ± 149.0	5%	278.2 ± 76.7	23%

## Data Availability

Not applicable.

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
