# Peer review of "Changes in Tumor Necrosis Factor α (TNFα) and Peptidyl Arginine Deiminase 4 (PAD-4) Levels in Serum of General Treated Psoriatic Patients"

_ijerph, 2022, doi:10.3390/ijerph19148723_

Round 1
Reviewer 1 Report
The manuscript has been improved as per the earlier suggestions and can be accepted for publications considering few minor concerns-
1. Cite some recent references of the year 2022.
2. Title of the manuscript should be simplified.
3. Abstract should be written more precisely.
Author Response
The manuscript has been improved as per the earlier suggestions and can be accepted for publications considering few minor concerns
We would like to thank to the Reviewers for all recommendations that help us to make our MS better. Thank you also for appreciating our efforts. We have responded to the criticisms below.
- Cite some recent references of the year 2022.
We would like to thank to the Reviewer for paying attention on the newest reports. In our MS, we have now 6 publication (both review as well as original paper) from 2022 (References: 45, 55-57 and 59-60).
- Title of the manuscript should be simplified.
The title has been changed.
- Abstract should be written more precisely.
Abstract part has been enriched with the details of the study.
Reviewer 2 Report
The authors of the article have made an honest attempt to answer the criticisms made of their study and have introdused improvements. So the article may be published.
Author Response
The authors of the article have made an honest attempt to answer the criticisms made of their study and have introdused improvements. So the article may be published.
We would like to thank to the Reviewers for all recommendations that help us to make our MS better. Thank you also for appreciating our efforts.
Reviewer 3 Report
Some English editing is required for this article.
Author Response
We would like to thank to the Reviewers for the recommendation. We have responded to the criticisms below.
Some English editing is required for this article.
Thank you very much for paying attention on the text quality. The entire text of the manuscript has been verified by native English-speaking editor. If some mistakes are still presented in our MS, we will verify them before publication with Editor from Journal.
This manuscript is a resubmission of an earlier submission. The following is a list of the peer review reports and author responses from that submission.
Round 1
Reviewer 1 Report
The authors have attempted well to work on the impact of biological therapies on TNFα and peptidyl arginine deiminase 4 levels in serum of patients with plaque psoriasis but the work seems to contain many flaws and the paper can not be accepted in its present form. My remarks for the paper have been presented below-
Minor comments-
- What was the criteria of choosing patient of Asia group more than 40 years?
- As title mention the impacted biological therapy then "what and why those biologics were selected for the treatment". Kindly mention the drug regimen profile.
- PASI stands for Psoriasis Area and Severity Index. Then why body surface area wasn't taken as variables? Do all the patient had same body surface?
Major Comments-
- As mentioned by you the duration of illness is 15 years. So, they all were receiving biologics for all duration or during the study. As long use of biologics cause immunosuppression.
- What were the drug regimens during the whole treatment period?
- The study was performed only in the group of 18 people, do think the results of the present study would be replicated with large sample size. I have a doubt, that the results of the study would significantly relate with large sample size.
Author Response
We thank the Reviewer for all valuable comments which were very helpful in making this MS better. All comments have been included to the new version of the MS.
Sincerely,
Joanna Czerwińska
Minor comments
- What was the criteria of choosing patient of Asia group more than 40 years?
The study included willing patients from Poland, aged 18-65 – without significant burdens and with min. 4 weeks break of treatment. The details about research groups have been included in the part: Material and methods of MS (p. 2, line 92).
- As title mention the impacted biological therapy then "what and why those biologics were selected for the treatment". Kindly mention the drug regimen profile.
In our Clinic, IL-17 and TNFα inhibitors are the most commonly used biological drugs, which is why we have the most patients treated with this therapy.
Patients were classified (depending on the symptoms) into four research groups. The first group used enstilar: externally for the first 4 weeks once a day, then for the next weeks, in proactive therapy - twice a week (maintenance treatment). The second group received an IL-17a inhibitor (secukinumab): subcutaneously, 300 mg at 0-4 weeks, then 300 mg every 1 month. The third therapy included the administration of anti-TNF (adalimumab): subcutaneously, initial dose 80 mg, after 7 days 40 mg every 2 weeks. The fourth group took methotrexate (MTX): orally, at a dose of 15 mg / week.
The details have been included to the part: Material and methods of manuscript (p. 3, lines: 97-104).
- PASI stands for Psoriasis Area and Severity Index. Then why body surface area wasn't taken as variables? Do all the patient had same body surface?
Initially, we only analyzed PASI score as a scale that assesses more variables in the disease: infiltration, scale, erythema, and body surface area, whereas BSA only assesses the body surface. In our Clinic, both PASI as well as BSA were taken as variables. In the presented version of MS, data has been added and Table 2 was modified (p.4, lines:129-136 and Discussion, p.7, line 210).
Major Comments
- As mentioned by you the duration of illness is 15 years. So, they all were receiving biologics for all duration or during the study. As long use of biologics cause immunosuppression.
All patients were treated with the selected method for the first time. The method was selected depending on the severity of the disease and the patient's history. We started each therapy and week 0 is our starting point. The typical therapy affects specific pathways in the pathomechanism of psoriasis – i.a. TNFα and anti-IL-17. This effect begins with the first administration of the drug.
- What were the drug regimens during the whole treatment period?
The treatment regimens have been added to the part: Material and methods of manuscript (p.3, lines: 97-104).
- The study was performed only in the group of 18 people, do think the results of the present study would be replicated with large sample size. I have a doubt, that the results of the study would significantly relate with large sample size.
We agree with the Reviewer that the research groups included in the MS are small. Currently, patients undergoing treatment at our Clinic and willing to participate in the study are constantly qualified for appropriate groups (in order to increase their number). The presented studies are pilot studies, the aim of which was to show whether the PAD-4 protein is present in the serum of psoriatic patients and healthy people, and whether the levels of this protein are subject to changes. In the literature, information that PAD-4 is not involved in the activation of NETs was found, so we aimed to check as it is in psoriasis. The obtained results are the basis for taking the next steps, i.e. research at the gene level, both its expression and sequence, as well as extended research at the protein level (expression and localization in lesional skin).
The part of above details have been included to the manuscript (Discussion, p., lines: 272-281).
Reviewer 2 Report
-you need to detail therapy groups in the text (not only in the explanation of figures)
-figures 1 and 2: not clear about which comparison the p value refers to
-line 127 & 136: there is no * in the figures !
-line 140: WHAT other cases ?!
-tables 3-5: statistically significant difference between WHICH time points: week 4 vs week 0, or week 12 vs week 0, or week 12 vs week 4 ?
-lines 170-172: interestingly, but not comparable: Bergen evaluated CLINICAL benefit, while you evaluated PAD-4 serum concentrations
-"Discussions" are mostly concentrated on NETs, rather then on the subject of your study (PAD-4)
Author Response
We thank the Reviewer for all valuable comments which were very helpful in making this MS better. The most of the comments have been included to the new version of the MS.
Sincerely,
Joanna Czerwińska
-you need to detail therapy groups in the text (not only in the explanation of figures)
We agree with the Reviewer that in the first version of our MS there was a lack of details about each therapy. The treatment regimens were added to the part: Material and methods of manuscript (p. 3, lines: 97-104).
-figures 1 and 2: not clear about which comparison the p value refers to
The p value refers to week 0 and week 12 – the first and last point of our measurements. We have added this information in figure captions.
-line 127 & 136: there is no * in the figures !
Asterisks have been added on each figure.
-line 140: WHAT other cases ?!
We apologize for unfortunate and reckless sentence. “In other cases” concerns (other than anti-TNFα therapy). We have changed it into “for other therapies”. (p.6, line 183)
-tables 3-5: statistically significant difference between WHICH time points: week 4 vs week 0, or week 12 vs week 0, or week 12 vs week 4 ?
Similarly like in case of figures: the p value refers to week 0 and week 12 – the first and last point of our measurements.
This information has been added in each description of figures and tables (Tables 3-5, Figures 1-2 ).
-lines 170-172: interestingly, but not comparable: Bergen evaluated CLINICAL benefit, while you evaluated PAD-4 serum concentrations
We agree with the Reviewer that the comparison was presented in an inappropriate way. It is clear that Bergen et al. evaluated clinical benefit, but in our study besides the serum level we also analyzed the PASI and BSA score. These results showed that the greatest improvement was achieved with therapy with anti-TNFα (PASI, BSA, PAD-4 level, TNFα level), but analyzing only PASI and BSA also anti-IL17A treatment was spectacular, what partially confirm the conclusion of Bergen et al.
The sentences have been modified (Discussion, p.7, lines: 213-218)
-"Discussions" are mostly concentrated on NETs, rather then on the subject of your study (PAD-4)
In the available literature, there is a lack of study analyzing PAD-4 protein concentration in human serum. Thus, the discussion about PAD-4 level is difficult. The information about PAD-4 is presented mainly in the context of NETs and still remains inconclusive.
The presented studies are pilot studies, the aim of which was to show whether the PAD-4 protein is present in the serum of psoriatic patients and healthy people, and whether the levels of this protein are subject to changes. In the literature, information that PAD-4 is not involved in the activation of NETs was found, so we aimed to check as it is in psoriasis. The obtained results are the basis for taking the next steps, i.e. research at the gene level, both its expression and sequence, as well as extended research at the protein level (expression and localization in lesional skin).
The part of above details have been included in the manuscript (Discussion, p.9, lines: 272-281). Additionally, we have added more information about NETs to the part Introduction (p.2, lines: 53-67) to connect the parts: Introduction and Discussion to improve the reception of the MS.
Reviewer 3 Report
The manuscript entitled ”The impact of biological therapies on tumor necrosis factor (TNFα) and peptidyl arginine deiminase 4 (PAD-4) levels in serum of patients with plaque psoriasis” by Joanna CzerwiÅ„ska et al focuses on determining PAD-4 and TNFα levels in serum of psoriatic patients and observing response of these factors to systemic (anti-17a, anti-TNFα and metotrexate) and local (enstilar) therapy. The increased levels of both PAD-4 and TNFα in pre-treatment patients have been reported and they were correlated with disease severity (PASI). It is one of the first studies of PAD-4 levels in serum of psoriatic patients. The authors claim that their results, showing the clear decrease in PAD-4 concentration, may indicate the enzyme involvement in the activation of NETs and psoriasis development. However, their conclusions are based on the results of only one ELISA method and using an extremely small number of patients.
The following improvements of the article would be useful for a better understanding of the impact of peptidyl arginine deiminase 4 (PAD-4) levels for psoriasis diagnostics and treatment:
1. The authors did not explain the benefits of determination of PAD-4 as an index of psoriasis severity.
2. Further exploration of the PAD-4 participation in psoriasis development is needed for revealing its significance.
3. The authors this article did not describe the involvement of overproduction of neutrophil extracellular traps in development of inflammatory cascade of psoriasis.
4. The authors’ conclusions made are doubtful because they used wary low patient sample, so the patient group for each type of therapy concluded 4-5 people. Besides the control group of healthy volunteers was half as much.
5. The authors this article used only one method ELISA for research. The work is based on the one method does not give the maneuver to discuss results.
6. It is necessary to supply materials with primary data.
Author Response
We thank the Reviewer for all valuable comments which were very helpful in making this MS better. The most of the comments have been included to the new version of the MS.
Sincerely,
Joanna Czerwińska
The following improvements of the article would be useful for a better understanding of the impact of peptidyl arginine deiminase 4 (PAD-4) levels for psoriasis diagnostics and treatment:
1. The authors did not explain the benefits of determination of PAD-4 as an index of psoriasis severity.
Fully defining PAD-4 as an indicator of psoriasis severity requires additional research at both the protein and the gene levels, but this could be an excellent new target for modern therapies, especially in patients exhibiting elevated levels of neutrophils. Therapy with PAD4 inhibitors may impede the functioning of the feedback loop between neutrophils (by stopping the NET formation), dendritic cells, lymphocytes and keratinocytes, and may become important in the treatment of autoimmune diseases.
The part of above details have been included to the manuscript (Discussion, p.9, lines: 272-281).
Further exploration of the PAD-4 participation in psoriasis development is needed for revealing its significance.
Discussing the role of PAD-4 in the pathogenesis of psoriasis is extremely difficult. In the literature, there are conflicting reports regarding the need to activate PAD-4 for the NET formation. However, there is no information on the contribution of PAD- 4 in psoriasis. Increased level of PAD-4 protein in the serum of patients with psoriasis and its decrease as a result of treatment may only suggest the participation of this enzyme in the activation of neutrophils to the formation of NETs. However, this hypothesis requires confirmation in studies of gene and protein expression in psoriatic plaques.
Part of the description is included in the part Introduction (p.2, lines: 66-86).
The authors this article did not describe the involvement of overproduction of neutrophil extracellular traps in development of inflammatory cascade of psoriasis.
The description has been added to the part Introduction (p.2, lines: 61-65) and Discussion (p.9, 272-281).
The authors’ conclusions made are doubtful because they used wary low patient sample, so the patient group for each type of therapy concluded 4-5 people. Besides the control group of healthy volunteers was half as much.
We agree with the Reviewer that the research groups included in the MS are small. Currently, patients undergoing treatment at our Clinic and willing to participate in the study are constantly qualified for appropriate groups (in order to increase their number). The presented studies are pilot studies, the aim of which was to show whether the PAD-4 protein is present in the serum of psoriatic patients and healthy people, and whether the levels of this protein are subject to changes. In the literature, information that PAD-4 is not involved in the activation of NETs was found, so we aimed to check as it is in psoriasis. The obtained results are the basis for taking the next steps, i.e. research at the gene level, both its expression and sequence, as well as extended research at the protein level (expression and localization in lesional skin).
The part of above details have been included to the manuscript (Discussion, p.9, lines: 272-281).
The authors this article used only one method ELISA for research. The work is based on the one method does not give the maneuver to discuss results.
We decided to start by analyzing the plasma levels of the PAD-4 protein to see if it was elevated in psoriasis patients. It was the fastest and relatively cheap test. The demonstration of fluctuations in PAD-4 levels is only an introduction to further studies of both gene and protein expression in the psoriatic skin. Further analysis is sure to provide more details on the role of PAD-4 in overproduction of NETs in patients with psoriasis.
It is necessary to supply materials with primary data.
In our opinion, all necessary data was presented in the tables
Reviewer 4 Report
1.Why did the authors link TNF-a and PAD-4? This should be described in Introduction.
2.The contribution of NETs to psoriasis pathogenesis is only partial. The authors think of its role in psoriasis too much exaggerated. If the authors have some evidence of the pathogenetic role of NETs in psoriasis, please show that.
3.I think that the contribution of PAD4 may be much more important in neutrophilic dermatosis such as Sweet' disease or pyoderma gangreosum. I recommend to apply this kind of correlation study in those disease above.
4. Though the authors lay stress on the correlation between PAD-4 and clinical scores of psoriasis, these may not be specific, just seeing the relations between CRP and psoriasis clinical indexes. If the authors consider NETs as the pathogenetic factor of psoriasis, please show the evidence in Introduction.
Author Response
We thank the Reviewer for all valuable comments which were very helpful in making this MS better. The most of the comments have been included to the new version of the MS.
Sincerely,
Joanna Czerwińska
1.Why did the authors link TNF-a and PAD-4? This should be described in Introduction.
Neutrophils may be activated in response to various microorganisms, such as bacteria, fungi and protozoa, but the induction of NETs does not need living pathogens and may be caused by various immune complexes, cytokines and chemokines. The main representative of the second group is considered TNFα. TNFα is a factor that activates PAD-4 enzyme, what was confirmed also by the correlation between both protein levels noted in our work. On the other hand, neutrophils are classified into two heterogenous populations: low-density granulocytes (LDGs) and typical polymorphonuclear neutrophils (PMNs). LDGs involved in NET creation also display an increased ability to synthesize TNFα.
The exploration has been added to the parts: Introduction (p.1-2, lines: 39-46).
2.The contribution of NETs to psoriasis pathogenesis is only partial. The authors think of its role in psoriasis too much exaggerated. If the authors have some evidence of the pathogenetic role of NETs in psoriasis, please show that.
We absolutely agree that pathogenesis of psoriasis is extremally complicated. The NET is only part of that process. The exploration has been added to the parts: Introduction (p.2, lines: 51-60).
3.I think that the contribution of PAD4 may be much more important in neutrophilic dermatosis such as Sweet' disease or pyoderma gangreosum. I recommend to apply this kind of correlation study in those disease above.
We are very grateful for this suggestion. We will certainly discuss our results in the context of the diseases mentioned above.
- Though the authors lay stress on the correlation between PAD-4 and clinical scores of psoriasis, these may not be specific, just seeing the relations between CRP and psoriasis clinical indexes. If the authors consider NETs as the pathogenetic factor of psoriasis, please show the evidence in Introduction.
The exploration has been added to the part Introduction (p.2, lines: 51-65, 74-79).
Round 2
Reviewer 3 Report
The authors have worked insufficiently to improve their manuscript.
Such flaws in the number of sample group for the experiments and a limited range of the methods used may threaten the validity of authors findings.
- In response to the first remark authors wrote: The part of above details has been included in the manuscript (Discussion, p.9, lines: 272-281). However, lines: 272-281 do not contain the Discussion and information in question.
- In response to the third remark authors of manuscript answered: The description has been added to the part Introduction (p.2, lines: 61-65) and Discussion (p.9, 272-281). However, the parts Introduction (p.2, lines: 61-65) and Discussion (p.9, 272-281) do not contain information about the involvement of overproduction of neutrophil extracellular traps in development of inflammatory cascade of psoriasis.
- In response the question authors wrote: The presented studies are pilot studies and the obtained results are the basis for taking the next steps, i.e. research at the gene level, both its expression and sequence, as well as extended research at the protein level.
The work will have scientific validity when the named researches will be performed with enuogh range of methods and number of sample groups.
In addition, the authors declined a request to provide supplementary material with primary data.
Author Response
We thank the Reviewer for an additional chance to improve our MS.
Sincerely,
Joanna Czerwińska
The authors have worked insufficiently to improve their manuscript. Such flaws in the number of sample group for the experiments and a limited range of the methods used may threaten the validity of authors findings.
In our opinion the basic conclusion from this article will not be changed. The PAD-4 level is elevated in serum of psoriatic patients in comparison to the control group (the differences are significant) and its level changes after cytokine-targeted therapies. This does not exclude the possibility of using the inhibitors of PAD-4 as a new therapy in treatment of psoriasis.
- In response to the first remark authors wrote: The part of above details has been included in the manuscript (Discussion, p.9, lines: 272-281). However, lines: 272-281 do not contain the Discussion and information in question.
We would like to apologize for a lack of that information in the uploaded version of MS. We have added the description to the MS (Introduction - detailed characteristics of the PAD isoforms, lines:85-105 and Discussion, lines: 302-331).
- In response to the third remark authors of manuscript answered: The description has been added to the part Introduction (p.2, lines: 61-65) and Discussion (p.9, 272-281). However, the parts Introduction (p.2, lines: 61-65) and Discussion (p.9, 272-281) do not contain information about the involvement of overproduction of neutrophil extracellular traps in development of inflammatory cascade of psoriasis.
We would like to apologize for a lack of description of the involvement of overproduction of neutrophil extracellular traps in development of inflammatory cascade of psoriasis. The description has been added to the part Introduction (lines: 64-84).
- In response the question authors wrote: The presented studies are pilot studies and the obtained results are the basis for taking the next steps, i.e. research at the gene level, both its expression and sequence, as well as extended research at the protein level. The work will have scientific validity when the named researches will be performed with enuogh range of methods and number of sample groups. In addition, the authors declined a request to provide supplementary material with primary data.
Our work presents the preliminary results, so we have decided to add the term “a pilot study” to the title to show that the project will be continued. In our opinion, additional tables with primary results will not be attractive to the readers. All important data was presented in the tables attached.
Reviewer 4 Report
The authors well addressed the issues I pointed out, and appropriately revised the manuscript. Although NETs is only one aspect of psoriasis pathogenesis, it may at least partially contribute to the development or exacerbation of psoriasis.
Author Response
The authors well addressed the issues I pointed out, and appropriately revised the manuscript. Although NETs is only one aspect of psoriasis pathogenesis, it may at least partially contribute to the development or exacerbation of psoriasis.
We would like to thank the Reviewer for that comment. We have added the information to the part Introduction.